# GLOBAL CONCAVITY AND OPTIMIZATION IN A CLASS OF DYNAMIC DISCRETE CHOICE MODELS

## ABSTRACT

Discrete choice models with unobserved heterogeneity are commonly used Econometric models for dynamic Economic behavior which have been adopted in practice to predict behavior of individuals and firms from schooling and job choices to strategic decisions in market competition. These models feature optimizing agents who choose among a finite set of options in a sequence of periods and receive choice-specific payoffs that depend on both variables that are observed by the agent and recorded in the data and variables that are only observed by the agent but not recorded in the data. Existing work in Econometrics assumes that optimizing agents are fully rational and requires finding a functional fixed point to find the optimal policy. We show that in an important class of discrete choice models the value function is globally concave in the policy. That means that simple algorithms that do not require fixed point computation, such as the policy gradient algorithm, globally converge to the optimal policy. This finding can both be used to relax behavioral assumption regarding the optimizing agents and to facilitate Econometric analysis of dynamic behavior. In particular, we demonstrate significant computational advantages in using a simple implementation policy gradient algorithm over existing "nested fixed point" algorithms used in Econometrics.

## 1 INTRODUCTION

Dynamic discrete choice model with unobserved heterogeneity is, arguably, the most popular model that is currently used for Econometric analysis of dynamic behavior of individuals and firms in Economics and Marketing (e.g. see surveys in Eckstein and Wolpin (1989), Dubé et al. (2002) Abbring and Heckman (2007), Aguirregabiria and Mira (2010)). Even most recent Econometric papers on single-agent dynamic decision-making use this setup to showcase their results (e.g. Arcidiacono and Miller, 2011; Aguirregabiria and Magesan, 2016; Müller and Reich, 2018).In this model, pioneered in Rust (1987), the agent chooses between a discrete set of options (typically 2) in a sequence of discrete time periods to maximize the expected cumulative discounted payoff. The reward in each period is a function of the state variable which follows a Markov process and is observed in the data and also a function of an idiosyncratic random variable that is only observed by the agent but is not reported in the data. The unobserved idiosyncratic component is designed to reflect heterogeneity of agents that may value the same choice differently.

Despite significant empirical success in prediction of dynamic economic behavior under uncertainty, dynamic discrete choice models frequently lead to seemingly unrealistic optimization problems that economic agents need to solve. For instance, Hendel and Nevo (2006) features an elaborate functional fixed point problem with constraints, which is computationally intensive, especially in continuous state spaces, for consumers to buy laundry detergent in the supermarket. Common approach for this functional fixed point problem is value function iteration (See Section 2.3 for more discussion).

At the same time, rich literature on Markov Decision Processes (cf. Sutton and Barto, 2018) have developed several effective optimization algorithms, such as the policy gradient algorithm and its variants, that do not require solving for a functional fixed point. However, the drawback of the policy gradient is that the value function in a generic Markov Decision problem is not concave in the policy. This means that gradient-based algorithms have no guarantees for global convergence for a generic MDP. While for some specific and simple models where closed-form characterizations exist, the

convergence results are shown by model-specific technique which is hard to generalize (e.g. Fazel et al., 2018, for linear quadratic regulator).

In this paper our main goal is to resolve the dichotomy in empirical social science literature that the rationality of consumers requires for them to be able to solve the functional fixed point problem which is computationally intensive. Our main theoretic contribution is the proof that, in the class of dynamic discrete choice models with unobserved heterogeneity, the value function of the optimizing agent is globally concave in the policy. This implies that a large set of policy gradient algorithms that have a modest computational power requirement for the optimizing agents have a fast convergence guarantee in our considered class of dynamic discrete choice models. The importance of this result is twofold.

First, it gives a promise that seemingly complicated dynamic optimization problems faced by consumers can be solved by relatively simple algorithms that do not require fixed point computation or functional optimization. This means that the policy gradient-style methods have an important *behavioral* interpretation. As a result, consumer behavior following policy gradient can serve as a behavioral assumption for estimating consumer preferences from data which is more natural for consumer choice settings than other assumptions that have been used in the past for estimation of preferences (e.g. $\epsilon$-regret learning in Nekipelov et al. (2015)). Second, more importantly, our result showing fast convergence of the policy gradient algorithm makes it an attractive alternative to the search for the functional fixed point in this class of problems. While the goal of the Econometric analysis of the data from dynamically optimizing consumers is to estimate consumer preferences by maximizing the likelihood function, it requires to sequentially solve the dynamic optimization problem for each value of utility parameters along the parameter search path. Existing work in Economics prescribes to use fixed point iterations for the value function to solve the dynamic optimization problem (see Rust (1987), Aguirregabiria and Mira (2007)). The replacement of the fixed point iterations with the policy gradient method significantly speeds up the maximization of the likelihood function. This makes the policy gradient algorithm our recommended approach for use in Econometric analysis, and establishes practical relevance of many newer reinforcement learning algorithms from behavioral perspective for social sciences.

## 2 Preliminaries

In this section, we introduce the concepts of the Markov decision process (MDP) with choice-specific payoff heterogeneity, the conditional choice probability (CCP) representation and the policy gradient algorithm.

### 2.1 Markov decision process

A discrete-time Markov decision process (MDP) with choice-specific heterogeneity is defined as a 5-tuple $\langle \mathcal{S}, \mathcal{A}, r, \epsilon, \mathcal{P}, \beta \rangle$, where $\mathcal{S}$ is compact convex state space with $\text{diam}(\mathcal{S}) \leq \tilde{S} < \infty$, $\mathcal{A}$ is the set of actions, $r : \mathcal{S} \times \mathcal{A} \to \mathbb{R}_+$ is the reward function, such that $r(s, a)$ is the immediate non-negative reward for the state-action pair $(s, a)$, $\epsilon$ are independent random variables, $\mathcal{P}$ is a Markov transition model where where $p(s'|s, a)$ defines the transition density between state $s$ and $s'$ under action $a$, and $\beta \in [0, 1)$ is the discount factor for future payoff. We assume that random variables $\epsilon$ are observed by the optimizing agent and not recorded in the data. These variables reflect idiosyncratic differences in preferences of different optimizing agents over choices. In the following discussion we refer to these variables as "random choice-specific shocks."

In each period $t = 1, 2, \ldots, \infty$, the nature realizes the current state $s_t$ based on the Markov transition $\mathcal{P}$ given the state-action pair $(s_{t-1}, a_{t-1})$ in the previous period $t-1$, and the choice-specific shocks $\epsilon_t = \{\epsilon_{t,a}\}_{a \in \mathcal{A}}$ drawn i.i.d. from distribution $\epsilon$. The optimizing agent chooses an action $a \in \mathcal{A}$, and her current period payoff is sum of the immediate reward and the choice-specific shock, i.e., $r(s, a) + \epsilon_{t,a}$. Given initial state $s_1$, the agent's long-term payoff is $\mathbf{E}_{\epsilon_1, s_2, \epsilon_2, \ldots} \left[ \sum_{t=1}^{\infty} \beta^{t-1} r(s_t, a_t) + \epsilon_{t,a_t} \right]$. This expression makes it clear that random shocks $\epsilon$ play a crucial role in this model by allowing us to define the *ex ante* value function of the optimizing agent which reflects the expected reward from agent's choices *before* the agent observes realization of $\epsilon_t$. When the distribution of shocks $\epsilon$ is sufficiently smooth (differentiable), the corresponding ex ante value function is smooth (differentiable)

as well. This allows us to characterize the impact of agent's policy on the expected value by considering functional derivatives of the value function with respect to the policy.

In the remainder of the paper, we rely on the following assumptions.

**Assumption 2.1.** *The state space $\mathcal{S}$ is compact in $\mathbb{R}$ and the action space $\mathcal{A}$ is binary, i.e., $\mathcal{A} = \{0, 1\}$.*

**Assumption 2.2.** *For all states $s$, the immediate reward $r(s, 0)$ for the state-action pair $(s, 0)$ is zero i.e., $r(s, 0) = 0$, and the immediate reward $r(s, 1)$ for the state-action pair $(s, 1)$ is bounded between $[R_{\min}, R_{\max}]$.*

**Assumption 2.3.** *Choice-specific shocks $\epsilon$ are Type I Extreme Value random variables with location parameter 0 (cf. Hotz and Miller, 1993) which are independent over choices and time periods.*

Assumption 2.1, 2.2, 2.3 are present in most of the papers on dynamic decision-making in economics, marketing and finance, (e.g. Dubé et al., 2002; Aguirregabiria and Mira, 2010; Arcidiacono and Miller, 2011; Aguirregabiria and Magesan, 2016; Müller and Reich, 2018)

**The policy and the value function** A stationary Markov policy is a function $\sigma : \mathcal{S} \times \mathbb{R}^{\mathcal{A}} \to \mathcal{A}$ which maps the current state $s$ and choice-specific shock $\epsilon$ to an action. In our further discussion we will show that there is a natural more restricted definition of the set of all feasible policies in this model.

Given any stationary Markov policy $\sigma$, the value function $V_{\sigma} : \mathcal{S} \to \mathbb{R}$ is a mapping from the initial state to the long-term payoff under policy $\sigma$, i.e.,

$$V_{\sigma}(s_1) = \mathbf{E}_{\epsilon_1, s_2, \epsilon_2, \ldots} \left[ \sum_{t=1}^{\infty} \beta^{t-1} \left\{ r(s_t, \sigma(s_t, \epsilon_t)) + \epsilon_{t, \sigma(s_t, \epsilon_t)} \right\} \right].$$

Since the reward is non-negative and bounded, and the discount $\beta \in [0, 1)$, value function $V_{\sigma}$ is well-defined and the optimal policy $\tilde{\sigma}$ (i.e., $V_{\tilde{\sigma}}(s) \geq V_{\sigma}(s)$ for all policies $\sigma$ and states $s$) exists. Furthermore, the following Bellman equation holds

$$V_{\sigma}(s) = \mathbf{E}_{\epsilon} \left[ r(s, \sigma(s, \epsilon)) + \epsilon_{\sigma(s, \epsilon)} + \beta \, \mathbf{E}_{s'} [V_{\sigma}(s') | s, \sigma(s, \epsilon)] \right] \qquad \text{for all policies } \sigma \qquad (1)$$

### 2.2 Conditional choice probability representation

Based on the Bellman equation (1) evaluated at the optimal policy, the optimal *Conditional Choice Probability* $\tilde{\delta}(a|s)$ (i.e., the probability of choosing action $a$ given state $s$ in the optimal policy $\tilde{\sigma}$) can be defined as

$$\tilde{\delta}(a|s) = \mathbf{E}_{\epsilon} [\mathbb{1}\{r(s, a) + \epsilon_a + \beta \, \mathbf{E}_{s'} [V_{\tilde{\sigma}}(s') | s, a] \geq r(s, a') + \epsilon_{a'} + \beta \, \mathbf{E}_{s'} [V_{\tilde{\sigma}}(s') | s, a'], \, \forall a'\}]$$

The optimal policy $\tilde{\sigma}$ can, therefore, be equivalently characterized by threshold function $\tilde{\pi}(s, a) = r(s, a) + \beta \, \mathbf{E}_{s'}[V_{\tilde{\sigma}}(s') | s, a]$, such that the optimizing agent chooses action $a^{\dagger}$ which maximizes the sum of the threshold and the choice-specific shock, i.e., $a^{\dagger} = \operatorname{argmax}_a \{\tilde{\pi}(s, a) + \epsilon_a\}$. Similarly, all non-optimal policies can be characterized by the corresponding threshold functions denoted $\pi$. Under Assumption 2.3 the conditional choice probability $\delta$ can be explicitly expressed in terms of the respective threshold $\pi$ as (cf. Rust, 1996)

$$\delta(a|s) = \exp(\pi(s, a)) \bigg/ \left( \sum_{a' \in \mathcal{A}} \exp(\pi(s, a')) \right).$$

We note that this expression induces a one-to-one mapping from the thresholds to the conditional choice probabilities. Therefore, all policies are fully characterized by their respective conditional choice probabilities. For notational simplicity, since we consider the binary action space $\mathcal{A} = \{0, 1\}$, and the reward $r(s, 0)$ is normalized to 0 we denote the immediate reward $r(s, 1)$ as $r(s)$; denote the conditional choice probability $\delta(0|s)$ as $\delta(s)$; and denote $\pi(s, 1)$ as $\pi(s)$.

In the subsequent discussion given that the characterization of policy $\sigma$ via its threshold is equivalent to its characterization by conditional choice probability $\delta$, we interchangeably refer to $\delta$ as the "policy." Then we rewrite the Bellman equation for a given policy $\delta$ as

$$\begin{aligned} V_{\delta}(s) = &(1 - \delta(s)) \, r(s) - \delta(s) \log(\delta(s)) \\ &- (1 - \delta(s)) \log(1 - \delta(s)) + \beta \, \mathbf{E}_{\epsilon, s'} \left[ V_{\delta}(s)(s') \Big| s \right] \end{aligned} \qquad (2)$$

Now we make two additional assumptions that are compatible with standard assumptions in the Econometrics literature.

**Assumption 2.4.** *For all states $s \in \mathcal{S}$, the conditional distribution of the next period Markov state $p(\cdot|s, 1)$ first-order stochastically dominates distribution $p(\cdot|s, 0)$, i.e., for all $\hat{s} \in \mathcal{S}$, $\mathbf{Pr}_{s'}[s' \leq \hat{s}|s, 1] \leq \mathbf{Pr}_{s'}[s' \leq \hat{s}|s, 0]$.*

**Assumption 2.5.** *Under the optimal policy $\tilde{\delta}$, the value function is non-decreasing in states, i.e., $V_{\tilde{\delta}}(s) \leq V_{\tilde{\delta}}(s')$ for all $s, s' \in \mathcal{S}$ s.t. $s < s'$.*

Consider a myopic policy $\bar{\delta}(s) = (\exp(r(s)) + 1)^{-1}$ which uses threshold $\bar{\pi}(s) = r(s)$. This policy corresponds to agent optimizing the immediate reward without considering how current actions impact future rewards. Under Assumption 2.4 and Assumption 2.5, the threshold for optimal policy is at least the threshold of myopic policy, i.e., $\tilde{\pi}(s) \geq \bar{\pi}(s)$. Hence, Lemma 2.1 holds.

**Lemma 2.1.** *The optimal policy $\tilde{\delta}$ chooses action 0 with weakly lower probability than the myopic policy $\bar{\delta}$ in all states $s \in \mathcal{S}$, i.e., $\tilde{\delta}(s) \leq \bar{\delta}(s)$.*

### 2.3 MDP in Economics and policy gradient

Our motivation in this paper comes from empirical work in Economics and Marketing where optimizing agents are consumers or small firms who make dynamic decisions while observing the current state $s$ and the reward $r(s, a)$ for their choice $a$. These agents often have limited computational power making it difficult for them to solve the Bellman equation to find the optimal policy. They also may have only sample access to the distribution of Markov transition which further complicates the computation of the optimal policy. In this context we contrast the value function iteration method which is based on solving the fixed point problem induced by the Bellman equation and the policy gradient method.

**Value function iteration** In the value function iterations, e.g., discussed in Jaksch et al. (2010); Haskell et al. (2016), the exact expectation in the Bellman equation (1) is replaced by an empirical estimate and then functional iteration uses the empirical Bellman equation to find the fixed point, i.e., the optimal policy. Under certain assumptions on MDPs, one can establish convergence guarantees for the value function iterations, e.g., Jaksch et al. (2010); Haskell et al. (2016). However, to run these iterations may require significant computation power which may not be practical when optimizing agents are consumers or small firms.

**Policy gradient** In contrast to value function iterations, policy gradient algorithm and its variations are model-free sample-based methods. At a high level, policy gradient parametrizes policies $\{\delta_\theta\}_{\theta \in \Theta}$ by $\theta \in \Theta$ and computes the gradient of the value function with respect to the current policy $\delta_\theta$ and update the policy in the direction of the gradient, i.e., $\theta \leftarrow \theta + \alpha \nabla_\theta V_{\delta_\theta}$. Though the individuals considered in the Economic MDP models may not compute the exact gradient with respect to a policy due to having only sample access to the Markov transition, previous work has provided approaches to produce an unbiased estimator of the gradient. For example, REINFORCE (Williams, 1992) updates the policy by $\theta \leftarrow \theta + \alpha R \nabla_\theta \log(\delta_\theta(a|s))$ where $R$ is the long-term payoff on path. Notice that this updating rule is simple comparing with value function iteration. The caveat of the policy gradient approach is the lack of its global convergence guarantee for a generic MDP. In this paper we show that such guarantee can be provided for the specific class of MDPs that we consider.

## 3 Warm-up: local concavity of the value function at the optimal policy

To understand the convergence of the policy gradient, in this section we introduce our main technique and show that the concavity of the value function with respect to policies is satisfied in a fixed neighborhood around the optimal policy. We rely on the special structure of the value function induced by random shocks $\epsilon$ which essentially "smooth it" making it differentiable. We then use Bellman equation (7) to compute strong Fréchet functional derivatives of the value functions and argue that the respective second derivative is negative at the optimal policy. We use this approach in Section 4 to show the global concavity of the value function with respect to policies.

By $\Delta$ we denote the convex compact set that contains all continuous functions $\delta : \mathcal{S} \to [0, 1]$ such that $0 \leq \delta(\cdot) \leq \bar{\delta}(\cdot)$. The Bellman equation (7) defines the functional $V_\delta(\cdot)$. Recall that Fréchet derivative of functional $V_\delta(\cdot)$, which maps bounded linear space $\Delta$ into the space of all continuous bounded functions of $s$, at a given $\delta(\cdot)$ is a bounded linear functional $\mathrm{D}V_\delta(\cdot)$ such that for all continuous $h(\cdot)$ with $\|h\|_2 \leq \bar{H}$: $V_{\delta+h}(\cdot) - V_\delta(\cdot) = \mathrm{D}V_\delta(\cdot)\, h(\cdot) + o(\|h\|_2)$. When functional $\mathrm{D}V_\delta(\cdot)$ is also Fréchet differentiable, we refer to its Fréchet derivative as the second Fréchet derivative of functional $V_\delta(\cdot)$ and denote it $\mathrm{D}^2 V_\delta(\cdot)$.

**Theorem 3.1.** *Value function $V_\delta$ is twice Freéchet differentiable with respect to $\delta$ at the choice probability $\tilde{\delta}$ corresponding to optimal policy and its Fréchet derivative is negative at $\tilde{\delta}$ in all states $s$, i.e., $\mathrm{D}^2 V_{\tilde{\delta}}(s) \leq 0$.*

We sketch the proof idea of Theorem 3.1 and defer its formal proof to Appendix A. Start with the Bellman equation (7) of the value function, the Fréchet derivative of the value function is the fixed point of the following Bellman equation

$$
\begin{aligned}
\mathrm{D}V_\delta(s) = {}& (\log(1 - \delta(s)) - \log(\delta(s)) - r(s)) \\
& + \beta\, (\mathbf{E}_{s'}[V_\delta(s')|s, 0] - \mathbf{E}_{s'}[V_\delta(s')|s, 1]) + \beta\, \mathbf{E}_{\epsilon, s'}[\mathrm{D}V_\delta(s')|s],
\end{aligned}
\tag{3}
$$

and

$$
\begin{aligned}
\mathrm{D}^2 V_\delta(s) = {}& -\frac{1}{\delta(s)(1 - \delta(s))} \\
& - 2\beta(\mathbf{E}_{s'}[\mathrm{D}V_\delta(s')|s, 1] - \mathbf{E}_{s'}[\mathrm{D}V_\delta(s')|s, 0]) + \beta\, \mathbf{E}_{s'}\left[\mathrm{D}^2 V_\delta(s')|s\right].
\end{aligned}
\tag{4}
$$

A necessary condition for its optimum yielding $\tilde{\delta}$ is $\mathrm{D}V_{\tilde{\delta}}(s) = 0$ for all states $s$. As a result, equation (9) implies that its second Fréchet derivative is negative for all states, i.e., $\mathrm{D}^2 V_{\tilde{\delta}}(s) \leq 0$.

The Bellman equation (9) of the second Fréchet derivative suggests that $\mathrm{D}^2 V_\delta(s) \leq 0$ for all states $s$ if

$$
\frac{1}{\delta(s)(1 - \delta(s))} + 2\beta(\mathbf{E}_{s'}[\mathrm{D}V_\delta(s')|s, 1] - \mathbf{E}_{s'}[\mathrm{D}V_\delta(s')|s, 0]) \geq 0
\tag{5}
$$

The first term in the inequality (5) is always positive for all policies in $\Delta$, but the second term can be arbitrary small. In the next section, we will introduce a nature smoothness assumption on MDP (i.e., Lipschitz MDP) and show that the local concavity can be extended to global concavity, which implies that the policy gradient algorithm for our problem converges globally under this assumption.

## 4 GLOBAL CONCAVITY OF THE VALUE FUNCTION

In this section, we introduce the notion of the Lipschitz Markov decision process, and Lipschitz policy space. We then restrict our attention to this subclass of MDPs. Our main result shows the optimal policy belongs to the Lipschitz policy space and the policy gradient globally converges in that space. We defer all the proofs of the results in this section to Appendix B.

### 4.1 LIPSCHITZ MARKOV DECISION PROCESS

Lipschitz Markov decision process has the property that for two state-action pairs that are close with respect to Euclidean metric in $\mathcal{S}$, their immediate rewards $r$ and Markovian transition $\mathcal{P}$ should be close with respect to the Kantorovich or $L_1$-Wasserstein metric. Kantorovich metric is, arguable, the most common metric used used in the analysis of MDPs (cf. Hinderer, 2005; Rachelson and Lagoudakis, 2010; Pirotta et al., 2015).

**Definition 4.1** (Kantorovich metric)**.** *For any two probability measures $p, q$, the Kantorovich metric between them is*

$$
\mathcal{K}(p, q) = \sup_f \left\{ \left| \int_X f\, d(p - q) \right| : f \text{ is 1-Lipschitz continuous} \right\}
$$

**Definition 4.2** (Lipschitz MDP)**.** *A Markov decision process is $(L_r, L_p)$-Lipschitz if*

$$
\begin{aligned}
& \forall s, s' \in \mathcal{S} && |r(s) - r(s')| \leq L_r\, |s - s'| \\
& \forall s, s' \in \mathcal{S}, a, a' \in \mathcal{A} && \mathcal{K}(p(\cdot|s, a), p(\cdot|s', a')) \leq L_p\, (|s - s'| + |a - a'|)
\end{aligned}
$$

## 4.2 CHARACTERIZATION OF THE OPTIMAL POLICY

Our result in Section 3, demonstrates that the second Fréchet derivative of $V_\delta$ with respect to $\delta$ is negative for a given policy $\delta$ when inequality (5) holds. To bound the second term of (5) from below, i.e., $\mathbf{E}_{s'}[\mathrm{D}V_\delta(s')|s,0] - \mathbf{E}_{s'}[\mathrm{D}V_\delta(s')|s,1]$, it is sufficient to show that Fréchet derivative $\mathrm{D}V_\delta(\cdot)$ is Lipschitz-continuous. Even though we already assume that the Markov transition is Lipschitz, it is still possible that $\mathrm{D}V_\delta$ is not Lipschitz: Bellman equation (8) for $\mathrm{D}V_\delta$ depends on policy $\delta(s)$ via $\log(1-\delta(s)) - \log(\delta(s))$, which can be non-Lipschitz in state $s$ for general policies $\delta$. Therefore, to guarantee Lipschitzness of the Fréchet derivative of the value function it is necessary to restrict attention to the space of Lipschitz policies. In this subsection, we show that this restriction is meaningful since the optimal policy is Lipschitz.

**Theorem 4.1.** *Given $(L_r, L_p)$-Lipschitz MDP, the optimal policy $\tilde{\delta}$ satisfies*

$$\left| \log\left( \frac{1-\tilde{\delta}(s)}{\tilde{\delta}(s)} \right) - \log\left( \frac{1-\tilde{\delta}(s^\dagger)}{\tilde{\delta}(s^\dagger)} \right) \right| \leq \left( L_r + \frac{2\beta R_{\max} L_p}{1-\beta} \right) |s - s^\dagger|$$

*for all state $s, s^\dagger \in \mathcal{S}$ where $R_{\max} = \max_{s\in\mathcal{S}} r(s)$ is the maximum of the immediate reward $r$ over $\mathcal{S}$.*

## 4.3 CONCAVITY OF THE VALUE FUNCTION WITH RESPECT TO LIPSCHITZ POLICIES

In this subsection, we present our main result showing the global concavity of the value function for our specific class of Lipschitz MDPs with unobserved heterogeneity over the space of Lipschitz policies.

**Definition 4.3.** *Given $(L_r, L_p)$-Lipschitz MDP, define its* Lipschitz policy space $\Delta$ *as*

$$\Delta = \{ \delta : \delta(s) \leq \bar{\delta}(s) \ \ \forall s \in \mathcal{S} \ \text{ and }$$

$$\left| \log\left( \frac{1-\delta(s)}{\delta(s)} \right) - \log\left( \frac{1-\delta(s^\dagger)}{\delta(s^\dagger)} \right) \right| \leq \left( L_r + \frac{2\beta R_{\max} L_p}{1-\beta} \right) |s - s^\dagger| \ \ \forall s, s^\dagger \in \mathcal{S} \Big\},$$

*where $\bar{\delta}$ is the myopic policy.*

Theorem 4.1 and Lemma 2.1 imply that the optimal policy $\tilde{\delta}$ lies in this Lipschitz policy space $\Delta$ for any Lipschitz MDP.

**Definition 4.4** (Condition for global convergence)**.** *We say that $(L_r, L_p)$-Lipschitz MDP satisfies the* sufficient *condition for global convergence if*

$$2\beta L_p < 1 \ \text{ and } \ \frac{2\beta L_p}{1-2\beta L_p}\left( 2L_r + \frac{4\beta R_{\max} L_p}{1-\beta} \right) \leq \frac{\left( \exp(R_{\min}) + 1 \right)^2}{\exp(R_{\min})}. \tag{6}$$

**Theorem 4.2.** *Given $(L_r, L_p)$-Lipschitz MDP which satisfies the condition for global convergence (6), value function $V_\delta$ is concave with respect to policy $\delta$ in the Lipschitz policy space $\Delta$, i.e., $\mathrm{D}^2 V_\delta(s) \leq 0$ for all $s \in \mathcal{S}, \delta \in \Delta$.*

## 4.4 THE RATE OF GLOBAL CONVERGENCE OF THE POLICY GRADIENT ALGORITHM

In this subsection, we establish the rate of global convergence a simple version of the policy gradient algorithm assuming oracle access to the Fréchet derivative of the value function. While this analysis provides only a theoretical guarantee, as discussed in Section 2.3, in practice the individuals are able to produce an unbiased estimator of the exact gradient. As a result, the practical application of the policy gradient algorithm would only need to adjust for the impact of stochastic noise in the estimator.

Since we assume that individuals know the immediate reward function $r$, the algorithm can be initialized at the myopic policy $\bar{\delta}$ with threshold $\bar{\pi}(s) = r(s)$, which is in the Lipschitz policy space $\Delta$. From Lemma 2.1 it follows that the myopic policy is pointwise in $\mathcal{S}$ greater than the optimal policy, i.e., $\bar{\delta}(s) \leq \tilde{\delta}(s)$. Consider policy $\underline{\delta}$ with threshold $\underline{\pi}(s) = r(s) + \frac{\beta}{1-\beta}R_{\max} - \frac{\beta}{2}R_{\min}$. Note that Bellman equation (7) implies that $V(s)$ is between $\frac{R_{\min}}{2}$ and $\frac{R_{\max}}{1-\beta}$ for all states $s$. Thus, policy $\underline{\delta}$ pointwise bounds the optimal policy $\tilde{\delta}$ from below, i.e., $\underline{\delta}(s) \leq \tilde{\delta}(s)$. Our convergence rate result applies to the policy gradient within the bounded Lipschitz policy set $\hat{\Delta}$.

**Definition 4.5.** *Given* $(L_r, L_p)$*-Lipschitz MDP, define its* bounded Lipschitz policy space $\hat{\Delta}$ *as*

$$\hat{\Delta} = \{\delta : \underline{\delta}(s) \leq \delta(s) \leq \bar{\delta}(s) \ \ \forall s \in \mathcal{S} \ \ and$$

$$\left| \log \left( \frac{1 - \delta(s)}{\delta(s)} \right) - \log \left( \frac{1 - \delta(s^\dagger)}{\delta(s^\dagger)} \right) \right| \leq \left( L_r + \frac{2\beta R_{\max} L_p}{1 - \beta} \right) \left| s - s^\dagger \right| \ \ \forall s, s^\dagger \in \mathcal{S} \right\}.$$

For simplicity of notation, we introduce constants $m$ and $M$ which only depend on $\beta$, $R_{\min}$, $R_{\max}$, $L_r$ and $L_p$, whose exact expressions are deferred to the supplementary material for this paper.

**Theorem 4.3.** *Given a* $(L_r, L_p)$*-Lipschitz MDP, which satisfies the condition for global convergence (6) and constants $m$ and $M$ defined above, for any step size $\alpha \leq \frac{1}{M}$, the policy gradient initialized at the myopic policy $\bar{\delta}$ and updating as $\delta \leftarrow \alpha \nabla_\delta V_\delta$ in the bounded Lipschitz policy space $\hat{\Delta}$ after $k$ iterations, it produces policy $\delta^{(k)}$ satisfying*

$$V_{\bar{\delta}}(s) - V_{\delta^{(k)}}(s) \leq \frac{(1 - \alpha m)^k}{(\exp(R_{\min}) + 1)^2} \qquad at \ all \ s \in \mathcal{S}.$$

## 5 Empirical application

To demonstrate the performance of the algorithm, we use the data from Rust (1987) which made the standard benchmark for the Econometric analysis of MDPs. The paper estimates the cost associated with maintaining and replacing bus engines using data from maintenance records from Madison Metropolitan Bus City Company over the course of 10 years (December, 1974—May, 1985). The data contains monthly observations on the mileage of each bus as well as the dates of major maintenance events (such as bus engine replacement).

Rust (1987) assumes that the engine replacement decisions follow an optimal stopping policy derived from solving a dynamic discrete choice model of the type that we described earlier. Using this assumption and the data, he estimates the cost of operating a bus as a function of the running mileage as well as the cost of replacing the bus engine. We use his estimates of the parameters of the return function and the state transition probabilities (bus mileage) to demonstrate convergence of the gradient descent algorithm.

In Rust (1987) the state $s_t$ is the running total mileage of the bus accumulated by the end of period $t$. The immediate reward is specified as a function of the running mileage as:

$$r(s_t, a, \theta_1) = \begin{cases} -\text{RC} + \epsilon_{t1}, & \text{if } a = 1 \\ -c(s_t, \theta_1) + \epsilon_{t0}, & \text{if } a = 0 \end{cases}$$

where RC is the cost of replacing the engine, $c(s_t, \theta_1)$ is the cost of operating a bus that has $s_t$ miles.

Following Rust (1987), we take $c(s_t, \theta_1) = \theta_1 s_t$. Further, as in the original paper, we discretize the mileage taking values in the range from 0 to 175 miles into an even grid of 2,571 intervals. Given the observed monthly mileage, Rust (1987) assumes that transitions on the grid can only be of increments $0, 1, 2, 3$ and $4$. Therefore, transition process for discretized mileage is fully specified by just four parameters $\theta_{2j} = \mathbf{Pr}[s_{t+1} = s_t + j | s_t, a = 0]$, $j = 0, 1, 2, 3$. Table 1 describes parameter values that we use directly from Rust (1987).

Table 1: parameter values in from Rust (1987).

| Parameter | Value |
| --- | --- |
| RC | 11.7257 |
| $\theta_1$ | $0.001 \times 2.45569$ |
| $(\theta_{20}, \theta_{21}, \theta_{22}, \theta_{23})$ | $(0.0937, 0.4475, 0.4459, 0.0127)$ |
| $\beta$ | 0.99 |

We use the gradient descent algorithm to update the policy threshold $\pi : \epsilon_1 + \pi \geq \epsilon_0 \Rightarrow a = 1$, where $a = 1$ denotes the decision to replace the engine. We set the learning rate using the RMSprop method[1].

---

[1]We use standard parameter values for RMSProp method: $\beta = 0.1$, $\nu = 0.001$ and $\epsilon = 10^{-8}$. The performance of the the method was very similar to that when we used ADAM to update the threshold values.

We use "the lazy projection" method to guarantee the search over Lipschitz policy space. The policy space is parametrized by the vector of thresholds $(\pi_1, \dots, \pi_N)$ corresponding to discretized state space $(s_1, \dots, s_N)$. It is initialized at the myopic policy, i.e. $\pi_1^{(0)} = u(s_1), \dots, \pi_N^{(0)} = u(s_N)$. At step $k$ the algorithm updates the thresholds to the value $\pi_i^{(k*)} = \pi_i^{(k-1)} - \alpha \mathrm{D}_{\delta^{(k-1)}} V(s_i) \mathcal{L}(\pi_i^{(k-1)})(1 - \mathcal{L}(\pi_i^{(k-1)}))$, where $\mathcal{L}(\cdot)$ is the logistic function and policy $\delta_j^{(k)} = \mathcal{L}(\pi_j^{(k-1)})$ for $i, j = 1, \dots, N$. To make the"lazy projection" updated values $\pi_i^{(k*)}$ are adjusted to the closest monotone set of values $\pi_1^{(k)} \leq \pi_2^{(k)} \leq \dots \leq \pi_N^{(k)}$. The algorithm terminates at step $k$ where the norm $\max_i |\mathrm{D}V_{\delta^{(k)}}(s_i)| \leq \tau$ for a given tolerance $\tau$.[2] The formal definition of lazy projection can be found in Appendix C.

Figure 3 demonstrates convergence properties of our considered version of the policy gradient algorithm. We used the "oracle" versions of the gradient and the value function that were obtained by solving the corresponding Bellman equations. We initialized the algorithm using the myopic threshold $\bar{\pi}(s) = -\mathrm{RC} + c(s, \theta_1)$; with the convergence criterion set to be based on the value $\max_i |\mathrm{D}V_\delta(s_i)|$.[3]

In the original model in Rust (1987), the discount factor used when estimating parameters of the cost function was very close to 1. However, performance of the algorithm improves drastically when the discount factor is reduced. This feature is closely related to the Hadamard stability of the solution of the Bellman equation (e.g. observed in Bajari et al. (2013)) and is not algorithm-specific. In all of the follow-up analysis by the same author (e.g. Rust (1996)) the discount factor is set to more moderate values of .99 or .9 indicating that these performance issues were indeed observed with the settings in Rust (1987). Figure 3 illustrates the performance of the algorithm for the case where the discount factor is set to $0.99$[4]. For the same convergence criterion, the algorithm converges much faster.

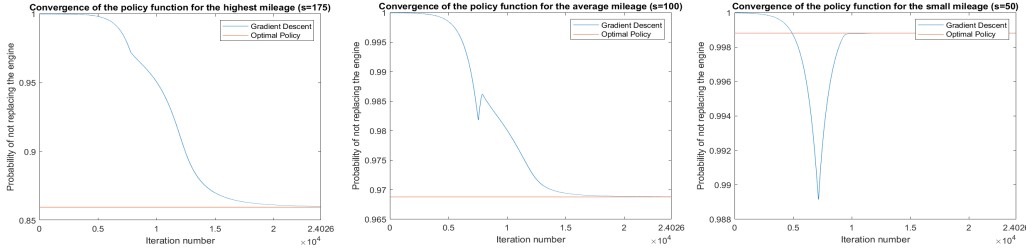

Figure 1: Convergence of gradient descent, discount factor $\beta = 0.99$

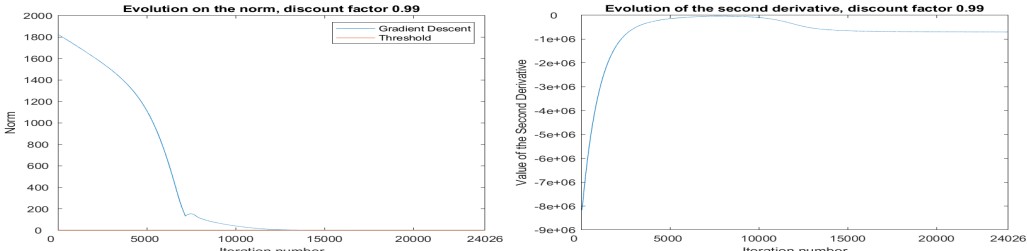

Figure 2: Performance of the norm $\max_i |\mathrm{D}V_\delta(s_i)|$ and the second derivative $\max_i |\mathrm{D}^2 V_\delta(s_i)|$, discount factor $\beta = 0.99$

---

[2]To optimize the performance of the method it is also possible to consider a mixed norm of the form $\max_i |\pi^{(k)}(s_i) - \pi^{(k-1)}(s_i)| + \lambda \max_i |DV_{\delta^{(k)}}(s_i)|_\infty \leq \tau$ for some calibrated weight $\lambda$. This choice would control both the rate of decay of the gradient and the advancement of the algorithm in adjusting the thresholds.

[3]The particular tolerance value used was 0.03 for illustrative purposes.

[4]When we reduce the cost of replacing the engine along with the discount factor, which ensures that there is significant variation in threshold values across states, convergence is improved even further

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

## APPENDIX

## A    OMITTED PROOF FOR THEOREM 3.1

**Theorem 3.1.** *Value function $V_\delta$ is twice Freéchet differentiable with respect to $\delta$ at the choice probability $\tilde{\delta}$ corresponding to optimal policy and its Fréchet derivative is negative at $\tilde{\delta}$ in all states $s$, i.e., $\mathrm{D}^2 V_{\tilde{\delta}}(s) \leq 0$.*

*Proof.* We start with the Bellman equation of the value function.

$$
\begin{aligned}
V_\delta(s) =& (1 - \delta(s))\, r(s) - \delta(s) \log(\delta(s)) \\
& - (1 - \delta(s)) \log(1 - \delta(s)) + \beta \, \mathbf{E}_{\epsilon, s'} \left[ V_\delta(s)(s') \Big| s \right]
\end{aligned}
\tag{7}
$$

First of all, note that in (7) the first three terms on the right hand side of the equation simple nonlinear functions $\delta(\cdot)$ and thus the directional derivative with respect to $\delta(\cdot)$ can be taken as an ordinary derivative with respect to $\delta$ as a parameter. Next note that if functional $J_\delta(\cdot)$ is directionally differentiable with respect to $\delta$ and for all $h(\cdot)$, $\frac{d}{d\tau} J_{\delta + \tau\, h}(\cdot)|_{\tau=0} / h(\cdot)$ is invariant, then $J_\delta(\cdot)$ is Fréchet differentiable with respect to $\delta$ and the obove ratio is its Fréchet derivative. As a result, the Fréchet derivative of simple functional $(1 - \delta(s))\, r(s) - \delta(s) \log(\delta(s)) - (1 - \delta(s)) \log(1 - \delta(s))$ with respect to $\delta(\cdot)$ exists and equal to $\log(1 - \delta(s)) - \log(\delta(s)) - r(s)$. This expression is itself a Freéchet-differentiable functional with Fréchet derivative equal to $-1/(\delta(s)(1 - \delta(s)))$, meaning that the original functional $(1 - \delta(s))\, r(s) - \delta(s) \log(\delta(s)) - (1 - \delta(s)) \log(1 - \delta(s))$ is twice Fréchet differentiable with the second Fréchet derivative $-1/(\delta(s)(1 - \delta(s)))$. Whenever the state transition is affected by the individual decision we need to consider decomposition of the conditional expectation with respect to the future state:

$$
\mathbf{E}_{\epsilon, s'}[V_\delta(s')|s] = (1 - \delta(s)) \, \mathbf{E}_{s'}[V_\delta(s')|s, 1] + \delta(s) \, \mathbf{E}_{s'}[V_\delta(s')|s, 0].
$$

Under standard technical conditions that allow the swap of the derivative and the integral

$$
\mathrm{D}\mathbf{E}_{s'}[V_\delta(s')|s] = (\mathbf{E}_{s'}[V_\delta(s')|s, 0] - \mathbf{E}_{s'}[V_\delta(s')|s, 1]) + \mathbf{E}_{s'}[\mathrm{D}V_\delta(s')|s]
$$

Thus, the Fréchet derivative of the value function should be the fixed point of the following Bellman equation

$$
\begin{aligned}
\mathrm{D}V_\delta(s) =& (\log(1 - \delta(s)) - \log(\delta(s)) - r(s)) \\
& + \beta \left( \mathbf{E}_{s'}[V_\delta(s')|s, 0] - \mathbf{E}_{s'}[V_\delta(s')|s, 1] \right) + \beta \, \mathbf{E}_{\epsilon, s'}[\mathrm{D}V_\delta(s')|s],
\end{aligned}
\tag{8}
$$

and

$$
\begin{aligned}
\mathrm{D}^2 V_\delta(s) =& -\frac{1}{\delta(s)(1 - \delta(s))} \\
& - 2\beta(\mathbf{E}_{s'}[\mathrm{D}V_\delta(s')|s, 1] - \mathbf{E}_{s'}[\mathrm{D}V_\delta(s')|s, 0]) + \beta \, \mathbf{E}_{s'}\left[\mathrm{D}^2 V_\delta(s')|s\right].
\end{aligned}
\tag{9}
$$

Given that both these equations are Type II Fredholm integral equations for $\mathrm{D}V_\delta(\cdot)$ and $\mathrm{D}^2 V_\delta(\cdot)$ which have unique solutions whenever $\beta < 1$ that are bounded and continuous (see Dunford and Schwartz (1957)) and, thus, unique solutions for both equations exist and $V_\delta(\cdot)$ is indeed Fréchet-differentiable. This means that the necessary condition for its optimum yielding $\tilde{\delta}$ is $\mathrm{D}V_{\tilde{\delta}}(s) = 0$ for all states $s$. As a result, equation (9) implies that its second Fréchet derivative is negative for all states, i.e., $\mathrm{D}^2 V_{\tilde{\delta}}(s) \leq 0$. $\qquad\square$

## B OMITTED PROOFS IN SECTION 4

### B.1 OMITTED PROOF OF THEOREM 4.1

**Theorem 4.1.** *Given an* $(L_r, L_p)$*-Lipschitz MDP, the optimal policy* $\tilde{\delta}$ *satisfies*

$$\left| \log\left( \frac{1 - \tilde{\delta}(s)}{\tilde{\delta}(s)} \right) - \log\left( \frac{1 - \tilde{\delta}(s^\dagger)}{\tilde{\delta}(s^\dagger)} \right) \right| \leq \left( L_r + \frac{2\beta R_{\max} L_p}{1 - \beta} \right) |s - s^\dagger|$$

*for all state* $s, s^\dagger \in \mathcal{S}$ *where* $R_{\max} = \max_{s \in \mathcal{S}} r(s)$ *is the maximum of the immediate reward* $r$ *over* $\mathcal{S}$.

*Proof.* At the optimal policy $\tilde{\delta}$, the Fréchet derivative of the value function is zero, i.e., $\mathrm{D}V_{\tilde{\delta}}(s) = 0$ for all state $s$. Therefore, from the Bellman equation (8) we establish that

$$\log\left( \frac{1 - \tilde{\delta}(s)}{\tilde{\delta}(s)} \right) = r(s) + \beta \left( \mathbf{E}_{s'} \left[ V_{\tilde{\delta}}(s') | s, 1 \right] - \mathbf{E}_{s'} \left[ V_{\tilde{\delta}}(s') | s, 0 \right] \right)$$

Thus, for all states $s, s^\dagger \in \mathcal{S}$,

$$\left| \mathbf{E}_{s'} \left[ V_{\tilde{\delta}}(s') | s, a \right] - \mathbf{E}_{s'} \left[ V_{\tilde{\delta}}(s') | s^\dagger, a \right] \right|$$

$$= \left| \int_{s' \in \mathcal{S}} V_{\tilde{\delta}}(s')(p(s'|s, a) - p(s'|s^\dagger, a)) ds' \right|$$

$$= \frac{R_{\max}}{1 - \beta} \left| \int_{s' \in \mathcal{S}} \frac{(1 - \beta)}{R_{\max}} V_{\tilde{\delta}}(s')(p(s'|s, a) - p(s'|s^\dagger, a)) ds' \right|$$

$$\leq \frac{R_{\max}}{1 - \beta} \sup_{\|f\|_L \leq 1} \left\{ \left| \int_{s' \in \mathcal{S}} f(s')(p(s'|s, a) - p(s'|s^\dagger, a)) ds' \right| \right\}$$

$$= \frac{R_{\max}}{1 - \beta} \mathcal{K}(p(\cdot|s, a), p(\cdot|s^\dagger, a)) \leq \frac{R_{\max} L_p}{1 - \beta} |s - s^\dagger|$$

where we use upper bounds $\sup_{s \in \mathcal{S}} V_{\tilde{\delta}}(s) \leq \frac{R_{\max}}{1 - \beta}$ and $\| \frac{(1 - \beta)}{R_{\max}} V_{\tilde{\delta}}(s') \|_L \leq 1$. Thus,

$$\left| \log\left( \frac{1 - \tilde{\delta}(s)}{\tilde{\delta}(s)} \right) - \log\left( \frac{1 - \tilde{\delta}(s')}{\tilde{\delta}(s^\dagger)} \right) \right|$$

$$= |r(s) - r(s^\dagger) + \beta \left( \mathbf{E}_{s'} \left[ V_{\tilde{\delta}}(s') | s, 1 \right] - \mathbf{E}_{s'} \left[ V_{\tilde{\delta}}(s') | s, 0 \right] \right)$$

$$\quad - \beta \left( \mathbf{E}_{s'} \left[ V_{\tilde{\delta}}(s') | s^\dagger, 1 \right] - \mathbf{E}_{s'} \left[ V_{\tilde{\delta}}(s') | s^\dagger, 0 \right] \right)|$$

$$\leq \left| r(s) - r(s^\dagger) \right| + \beta \left| \mathbf{E}_{s'} \left[ V_{\tilde{\delta}}(s') | s, 1 \right] - \mathbf{E}_{s'} \left[ V_{\tilde{\delta}}(s') | s^\dagger, 1 \right] \right|$$

$$\quad + \beta \left| \mathbf{E}_{s'} \left[ V_{\tilde{\delta}}(s') | s, 0 \right] - \mathbf{E}_{s'} \left[ V_{\tilde{\delta}}(s') | s^\dagger, 0 \right] \right|$$

$$\leq \left( L_r + \frac{2\beta R_{\max} L_p}{1 - \beta} \right) |s - s^\dagger| \qquad \qquad \square$$

### B.2 OMITTED PROOF OF THEOREM 4.2

**Theorem 4.2.** *Given an* $(L_r, L_p)$*-Lipschitz MDP which satisfies the condition for global convergence (6), the value function* $V_\delta$ *is concave with respect to policy* $\delta$ *in the Lipschitz policy space* $\Delta$, *i.e.,* $\mathrm{D}^2 V_\delta(s) \leq 0$ *for all* $s \in \mathcal{S}$, $\delta \in \Delta$.

To show Theorem 4.2, we first introduce the following lemma establishing Lipschitz continuity of the Fréchet derivative of the value function.

**Lemma B.1.** *Given a* $(L_r, L_p)$*-Lipschitz MDP, for all policies* $\delta$ *in the the Lipschitz policy space* $\Delta$, *the Fréchet derivative of the respective value function* $\mathrm{D}V_\delta(\cdot)$ *is* $\left( \frac{2L_r + \frac{4\beta R_{\max} L_p}{1 - \beta}}{1 - 2\beta L_p} \right)$*-Lipschitz*

*continuous, i.e., for all states $s, s^\dagger \in \mathcal{S}$,*

$$\left| \mathrm{DV}_\delta(s) - \mathrm{DV}_\delta(s^\dagger) \right| \leq \left( \frac{2L_r + \frac{4\beta R_{\max} L_p}{1-\beta}}{1 - 2\beta L_p} \right) \left| s - s^\dagger \right|.$$

*Proof.* We begin with the Bellman equation (8) for the Fréchet derivative of value function.

$$\mathrm{DV}_\delta(s) = \log\left( \frac{1 - \delta(s)}{\delta(s)} \right) - r(s)$$
$$+ \beta \left( \mathbf{E}_{s'}[V_\delta(s')|s, 0] - \mathbf{E}_{s'}[V_\delta(s')|s, 1] \right) + \beta \mathbf{E}_{\epsilon, s'}[\mathrm{DV}_\delta(s')|s]$$

We use the concept of the contraction mapping to prove the result of the Lemma.

**Definition B.1.** *Let $T : X \to X$ be a mapping from a metric space $X$ to itself,*

- *$T$ is a* contraction mapping *(with modulus $\gamma \in [0, 1)$) if $\rho(T(x), T(y)) \leq \gamma \rho(x, y)$ for all $x, y \in X$, where $\rho$ is a metric on $X$.*

- *$x$ is a* fixed point *of $T$ if $T(x) = x$.*

**Lemma B.2.** *Suppose that $X$ is a complete metric space and that $T : X \to X$ is a contraction mapping with modulus $\gamma$. Then,*

- *$T$ has a unique fixed point $x^*$.*

- *If $X' \subseteq X$ is a closed subset for which $T(X') \subseteq X'$, then $x^* \in X'$.*

Consider the contraction mapping $T_\delta(x)(s) = \log\left( \frac{1-\delta(s)}{\delta(s)} \right) - r(s) + \beta \left( \mathbf{E}_{s'}[V_\delta(s')|s, 0] - \mathbf{E}_{s'}[V_\delta(s')|s, 1] \right) + \beta \mathbf{E}_{\epsilon, s'}[x(s')|s]$, then the Bellman equation implies that $\mathrm{DV}_\delta$ is the fixed point of contraction mapping $T_\delta$. Since the Lipschitz continuity property forms a closed subset, by Lemma B.2, it is sufficient to show for any $L_{DV}$-Lipschitz continuous $x$, $T_\delta(x)$ is also $L_{DV}$-Lipschitz continuous, where $L_{DV} = \frac{2L_r + \frac{4\beta R_{\max} L_p}{1-\beta}}{1 - 2\beta L_p}$. Thus, consider states $s, s^\dagger \in \mathcal{S}$,

$$\left| T(x)(s) - T(x)(s^\dagger) \right|$$
$$\leq \left| \log\left( \frac{1 - \delta(s)}{\delta(s)} \right) - \log\left( \frac{1 - \delta(s^\dagger)}{\delta(s^\dagger)} \right) \right| + \left| r(s) - r(s^\dagger) \right|$$
$$+ \beta \left| \mathbf{E}_{s'}[V_\delta(s')|s, 0] - \mathbf{E}_{s'}\left[V_\delta(s')|s^\dagger, 0\right] \right|$$
$$+ \beta \left| \mathbf{E}_{s'}[V_\delta(s')|s, 1] - \mathbf{E}_{s'}\left[V_\delta(s')|s^\dagger, 1\right] \right|$$
$$+ \beta \left| \mathbf{E}_{s'}[x(s')|s, 0]\,\delta(s) - \mathbf{E}_{s'}\left[x(s')|s^\dagger, 0\right]\delta(s^\dagger) \right|$$
$$+ \beta \left| \mathbf{E}_{s'}[x(s')|s, 1]\,(1 - \delta(s)) - \mathbf{E}_{s'}\left[x(s')|s^\dagger, 1\right](1 - \delta(s^\dagger)) \right|$$

where

$$\left| \log\left( \frac{1 - \delta(s)}{\delta(s)} \right) - \log\left( \frac{1 - \delta(s^\dagger)}{\delta(s^\dagger)} \right) \right| \leq \left( L_r + \frac{2\beta R_{\max} L_p}{1 - \beta} \right) \left| s - s^\dagger \right|$$
$$\left| r(s) - r(s^\dagger) \right| \leq L_r \left| s - s^\dagger \right|$$

by the same calculation in the proof of Theorem 4.1, for $a = 0, 1$,

$$\beta \left| \mathbf{E}_{s'}[V_\delta(s')|s, a] - \mathbf{E}_{s'}\left[V_\delta(s')|s^\dagger, a\right] \right| \leq \left( L_r + \frac{2\beta R_{\max} L_p}{1 - \beta} \right) \left| s - s^\dagger \right|$$

and

$$
\left| \mathbf{E}_{s'}\left[ x(s') | s, 0 \right] \delta(s) - \mathbf{E}_{s'}\left[ x(s') | s^{\dagger}, 0 \right] \delta(s^{\dagger}) \right|
$$

$$
= \left| \int_{s' \in \mathcal{S}} (\delta(s) - \delta(s^{\dagger})) x(s') (p(s'|s,a) - p(s'|s^{\dagger},a)) ds' \right|
$$

$$
= L_{DV} \left| \int_{s' \in \mathcal{S}} \frac{(\delta(s) - \delta(s^{\dagger})) x(s')}{L_{DV}} (p(s'|s,a) - p(s'|s^{\dagger},a)) ds' \right|
$$

$$
\leq L_{DV} \sup_{\|f\|_L \leq 1} \left\{ \left| \int_{s' \in \mathcal{S}} f(s') (p(s'|s,a) - p(s'|s^{\dagger},a)) ds' \right| \right\}
$$

$$
= L_{DV} \mathcal{K}(p(\cdot|s,a), p(\cdot|s^{\dagger},a)) \leq L_{DV} L_p \left| s - s^{\dagger} \right|
$$

where we use the bound $\left| \delta(s) - \delta(s^{\dagger}) \right| \leq 1$ and thus $\| \frac{(\delta(s) - \delta(s^{\dagger})) x(s')}{L_{DV}} \|_L \leq 1$. Similarly,

$$
\left| \mathbf{E}_{s'}\left[ x(s')|s, 1 \right] (1 - \delta(s)) - \mathbf{E}_{s'}\left[ x(s')|s^{\dagger}, 1 \right] (1 - \delta(s^{\dagger})) \right| \leq L_{DV} L_p \left| s - s^{\dagger} \right|
$$

Combining all the bounds, we obtain that

$$
\left| T(x)(s) - T(x)(s^{\dagger}) \right| \leq \left( 2L_r + \frac{4\beta R_{\max} L_p}{1 - \beta} + 2\beta L_{DV} L_p \right) \left| s - s^{\dagger} \right|.
$$

Substitution $L_{DV} = \frac{2L_r + \frac{4\beta R_{\max} L_p}{1 - \beta}}{1 - 2\beta L_p}$ yields the statement of the Lemma. $\qquad \square$

*Proof of Theorem 4.2.* From the Bellman equation (9), it is sufficient to show

$$
\frac{1}{\delta(s)(1 - \delta(s))} \geq 2\beta (\mathbf{E}_{s'}[DV_\delta(s')|s, 1] - \mathbf{E}_{s'}[DV_\delta(s')|s, 0]) \tag{10}
$$

We bound both sides separately. Since the policy satisfies $\delta(s) \leq \bar{\delta}(s)$ for all states $s$, and $\bar{\delta}(s) = \frac{1}{\exp(r(s)) + 1} \leq \frac{1}{2}$, the left hand side can be bounded from below as

$$
\frac{1}{\delta(s)(1 - \delta(s))} \geq \frac{1}{\bar{\delta}(s)(1 - \bar{\delta}(s))} \geq \frac{\left( \exp(R_{\min}) + 1 \right)^2}{\exp(R_{\min})}
$$

Meanwhile, the righthand side can be bounded from above by Lemma B.1. Let $L_{DV} = \frac{2L_r + \frac{4\beta R_{\max} L_p}{1 - \beta}}{1 - 2\beta L_p}$,

$$
2\beta \left| \mathbf{E}_{s'}[DV_\delta(s')|s, 1] - \mathbf{E}_{s'}[DV_\delta(s')|s, 0] \right|
$$

$$
= 2\beta \left| \int_{s' \in \mathcal{S}} DV_\delta(s')(p(s'|s, 1) - p(s'|s, 0)) ds' \right|
$$

$$
= 2\beta L_{DV} \left| \int_{s' \in \mathcal{S}} \frac{DV_\delta(s')}{L_{DV}} (p(s'|s, 1) - p(s'|s, 0)) ds' \right|
$$

$$
\leq 2\beta L_{DV} \sup_{\|f\|_L \leq 1} \left\{ \left| \int_{s' \in \mathcal{S}} f(s')(p(s'|s, 1) - p(s'|s, 0)) ds' \right| \right\}
$$

$$
= 2\beta L_{DV} \mathcal{K}(p(\cdot|s, 1), p(\cdot|s, 0)) \leq \frac{2\beta L_p}{1 - 2\beta L_p} \left( 2L_r + \frac{4\beta R_{\max} L_p}{1 - \beta} \right)
$$

From the condition of global convergence

$$
\frac{2\beta L_p}{1 - 2\beta L_p} \left( 2L_r + \frac{4\beta R_{\max} L_p}{1 - \beta} \right) \leq \frac{\left( \exp(R_{\min}) + 1 \right)^2}{\exp(R_{\min})}
$$

it follows that the inequality (10) is satisfied and the Bellman equation (9) implies that $D^2 V_\delta(s) \leq 0$ for all states $s \in \mathcal{S}$. $\qquad \square$

### B.3  OMITTED PROOF OF THEOREM 4.3

For notation simplicity, we introduce notations $m$ and $M$ such that

$$m = \frac{1}{1-\beta}\left(\frac{\left(\exp(R_{\min})+1\right)^2}{\exp(R_{\min})} - \frac{2\beta L_p}{1-2\beta L_p}\left(2L_r + \frac{4\beta R_{\max}L_p}{1-\beta}\right)\right)$$

$$M = \frac{1}{(1-\beta)^2}\left((1-\beta)\frac{\left(\exp\left(\frac{1}{1-\beta}R_{\max} - \frac{\beta}{2}R_{\min}\right)+1\right)^2}{\exp(\frac{1}{1-\beta}R_{\max} - \frac{\beta}{2}R_{\min})}\right.$$
$$\left. + 2\beta\left(\exp\left(\frac{1}{1-\beta}R_{\max} - \frac{\beta}{2}R_{\min}\right) + \frac{2\beta}{1-\beta}R_{\max} - (1+\beta)R_{\min}\right)\right)$$

**Theorem 4.3.** *Given a $(L_r, L_p)$-Lipschitz MDP, which satisfies the condition for global convergence (6) and constants $m$ and $M$ defined above, for any step size $\alpha \leq \frac{1}{M}$, the policy gradient initialized at the myopic policy $\bar{\delta}$ and updating as $\delta \leftarrow \alpha \nabla_\delta V_\delta$ in the bounded Lipschitz policy space $\hat{\Delta}$ after $k$ iterations, it produces policy $\delta^{(k)}$ satisfying*

$$V_{\bar{\delta}}(s) - V_{\delta^{(k)}}(s) \leq \frac{(1-\alpha m)^k}{(\exp(R_{\min})+1)^2}$$

*at all $s \in \mathcal{S}$.*

Our analysis follows the standard steps establishing convergence of the conventional gradient descent algorithm which bounds the second Fréchet derivative of the value function $V_\delta$ with respect to the policy $\delta$ from above and from below by $m$ and $M$ respectively.

**Lemma B.3.** *Given a $(L_r, L_p)$-Lipschitz MDP, which satisfies the condition for global convergence (6), for all policies $\delta$ in the bounded Lipschitz policy space $\hat{\Delta}$, for all states $s \in \mathcal{S}$, the second Fréchet derivative of the value function $V_\delta$ with respect to the policy $\delta$ is upperbounded as*

$$\mathrm{D}^2 V_\delta(s) \leq -m.$$

*Proof.* The Bellman equation (9) implies that

$$\max_s \mathrm{D}^2 V_\delta(s) \leq \frac{1}{1-\beta}\left(-\min_s \frac{1}{\delta(s)(1-\delta(s))}\right.$$
$$\left. + 2\beta\max_s(\mathbf{E}_{s'}[\mathrm{D}V_\delta(s')|s,0] - \mathbf{E}_{s'}[\mathrm{D}V_\delta(s')|s,1])\right)$$

By the same argument as in Theorem 4.2,

$$\min_s \frac{1}{\delta(s)(1-\delta(s))} \geq \frac{\left(\exp(R_{\min})+1\right)^2}{\exp(R_{\min})}$$

$$\max_s(\mathbf{E}_{s'}[\mathrm{D}V_\delta(s')|s,1] - \mathbf{E}_{s'}[\mathrm{D}V_\delta(s')|s^\dagger,0]) \leq \frac{2\beta L_p}{1-2\beta L_p}\left(2L_r + \frac{4\beta R_{\max}L_p}{1-\beta}\right)$$

Thus, for all state $s \in \mathcal{S}$,

$$\mathrm{D}^2 V_\delta(s) \leq -m. \quad \square$$

**Lemma B.4.** *Given a $(L_r, L_p)$-Lipschitz MDP, which satisfies the condition for global convergence, for all policy $\delta$ in the bounded Lipschitz policy space $\hat{\Delta}$, for all state $s \in \mathcal{S}$, the second derivative of the value function $V_\delta$ with respect to the policy $\delta$ is is lowerbounded as*

$$\mathrm{D}^2 V_\delta(s) \geq -M.$$

*Proof.* The Bellman equation (9) implies that

$$\min_s \mathrm{D}^2 V_\delta(s) \geq \frac{1}{1-\beta}\left(-\max_s \frac{1}{\delta(s)(1-\delta(s))}\right.$$
$$\left. + 2\beta\left(\min_s \mathrm{D}V_\delta(s) - \max_s \mathrm{D}V_\delta(s)\right)\right)$$

By restricting policy to the bounded Lipschitz policy space $\hat{\Delta}$ we bound

$$\max_s \frac{1}{\delta(s)(1-\delta(s))} \leq \max_s \frac{1}{\underline{\delta}(s)(1-\underline{\delta}(s))} \leq \frac{\left(\exp\left(\frac{1}{1-\beta}R_{\max} - \frac{\beta}{2}R_{\min}\right)+1\right)^2}{\exp\left(\frac{1}{1-\beta}R_{\max} - \frac{\beta}{2}R_{\min}\right)}$$

Provided

$$\min_s V_\delta(s) \geq \frac{R_{\min}}{2}$$

$$\max_s V_\delta(s) \leq \frac{R_{\max}}{1-\beta}$$

$$\min_s \left(\log\left(\frac{1-\delta(s)}{\delta(s)}\right) - r(s)\right) \geq \min_s \left(\log\left(\frac{1-\bar{\delta}(s)}{\bar{\delta}(s)}\right) - r(s)\right) = 0$$

$$\max_s \left(\log\left(\frac{1-\delta(s)}{\delta(s)}\right) - r(s)\right) \leq \max_s \log\left(\frac{1-\underline{\delta}(s)}{\underline{\delta}(s)}\right) - \min_s r(s)$$

$$\leq \exp\left(\frac{1}{1-\beta}R_{\max} - \frac{\beta}{2}R_{\min}\right) - R_{\min}$$

it follows from Bellman equation (8) that

$$\min_s DV_\delta(s) \geq \frac{1}{1-\beta}\left(\min_s\left(\log\left(\frac{1-\delta(s)}{\delta(s)}\right) - r(s)\right) + \beta(\min_s V_\delta(s) - \max_s V_\delta(s))\right)$$

$$\geq \frac{\beta}{1-\beta}\left(\frac{R_{\min}}{2} - \frac{R_{\max}}{1-\beta}\right)$$

$$\max_s DV_\delta(s) \leq \frac{1}{1-\beta}\left(\max_s\left(\log\left(\frac{1-\delta(s)}{\delta(s)}\right) - r(s)\right) + \beta(\max_s V_\delta(s) - \min_s V_\delta(s))\right)$$

$$\leq \frac{1}{1-\beta}\left(\exp\left(\frac{1}{1-\beta}R_{\max} - \frac{\beta}{2}R_{\min}\right) + \frac{\beta}{1-\beta}R_{\max} - \frac{2+\beta}{2}R_{\min}\right)$$

Thus, for all state $s \in \mathcal{S}$,

$$D^2 V_\delta(s) \geq -M. \quad \square$$

*Proof of Theorem 4.3.* The convergence rate guarantee follows from Lemma B.3 and Lemma B.4, under the standard arguments for the gradient descent algorithm for $m$-strongly concave and $M$-smooth (i.e., $M$-Lipschitz gradient) functions (cf. Bansal and Gupta, 2017). $\quad \square$

## C   MORE RESULTS IN SECTION 5

---

**Algorithm 1** "Lazy projection", $(\pi_1, \ldots, \pi_N)$: thresholds corresponding to discretized state space $(s_1, \ldots, s_N)$; $\mathcal{L}(\cdot)$: logistic function; policy $\delta_j = \mathcal{L}(\pi_j)$; $\alpha$: step size; $\tau$: termination tolerance

---

1: $\pi_1^{(0)} \leftarrow u(s_1), \ldots, \pi_N^{(0)} \leftarrow u(s_N)$ // Initialize $\pi^{(}0)$ at the myopic policy
2: **while** $\max_i |DV_{\delta^{(k)}}(s_i)| \leq \tau$ **do**
3: $\quad \pi_i^{(k*)} \leftarrow \pi_i^{(k-1)} - \alpha\, D_{\delta^{(k-1)}} V_{\delta^{(k-1)}}(s_i)\mathcal{L}(\pi_i^{(k-1)})(1 - \mathcal{L}(\pi_i^{(k-1)}))$ *for all* $i \in [N]$
4: $\quad (\pi_1^{(k)}, \ldots, \pi_N^{(k)}) \leftarrow$ the closest monotone thresholds of $(\pi_1^{(k*)}, \ldots, \pi_N^{(k*)})$ // Lazy projection
5: **return** $(\pi_1^{(k)}, \ldots, \pi_N^{(k)})$

---

We list the convergence of gradient descent and its derivative, second derivative at smaller discount factor $\beta = 0.9$.

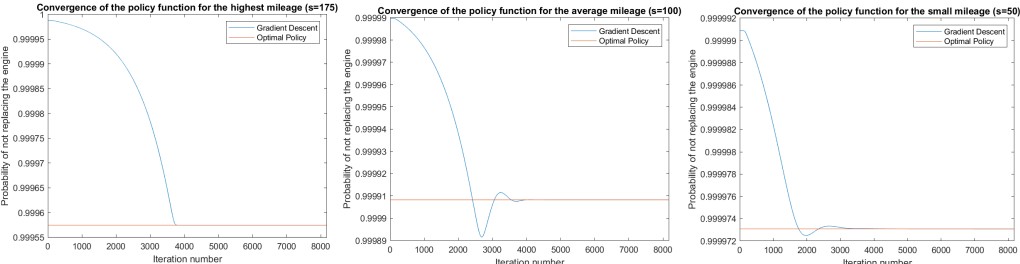

Figure 3: Convergence of gradient descent, discount factor $\beta = 0.9$

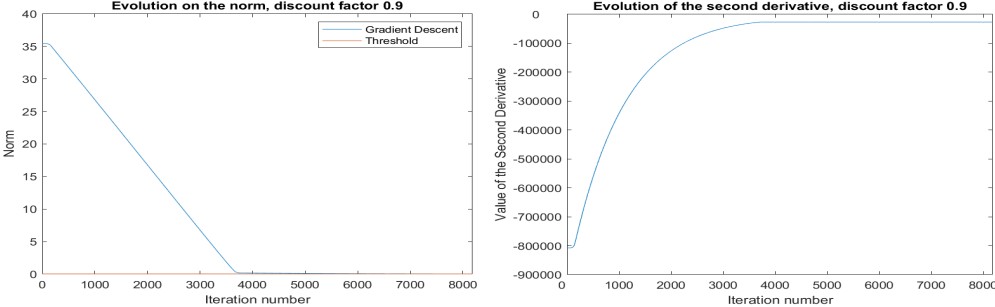

Figure 4: Performance of the norm $\max_i |\mathrm{D}V_\delta(s_i)|$ and the second derivative $\max_i |\mathrm{D}^2 V_\delta(s_i)|$, discount factor $\beta = 0.9$

