# OpenReview forum: "Global Concavity and Optimization in a Class of Dynamic Discrete Choice Models"
_ICLR.cc/2020/Conference — Reject_

### Official Review · AnonReviewer1 · 2019-10-22
**Official Blind Review #1**

**Rating:** 3

**Review:**

This paper considers reinforcement learning for discrete choice models with unobserved heterogeneity, which is useful for analyzing dynamic Economic behavior.  Random choice-specific shocks in reward is accommodated, which are only observed by the agent but not recorded in the data. Existing optimization approaches rely on finding a functional fixed point, which is computationally expensive.  The main contribution of the paper lies in formulating discrete choice models into an MDP, and showing that the value function is concave with respect to the policy (represented by conditional choice probability).  So policy gradient algorithm can provably converge to the global optimal.  Conditions on the parameters for global concavity are identified and rates of convergences are established.  Finally, significant advantages in computation were demonstrated on the data from Rust (1987), compared with “nested fixed point” algorithms that is commonly used in Econometrics.

This paper is well written.  The most important and novel result is the concavity of the value function with respect to the policy.  My major concerns are:

1. How restrictive the assumptions are in Definition 1.4?  In particular, R_min is defined from Assumption 2.2 as “the immediate reward … is bounded between [R_min, R_max]”.  So if we set R_min to negative infinity, the right-hand side of Eq 6 will be infinity, and so the condition is always met.  Is this really true?  At least, does the experiment satisfy Definition 1.4?

2. The experiment is on a relatively small problem.  Solving value/policy iteration with 2571 states and 2 actions is really not so hard, and many efficient algorithms exist other than value/policy iteration.  For example, a variant of policy iteration where policy evaluation is not solved exactly, but instead approximated by applying a small number of Bellman iterations. Or directly optimize the Bellman residual by, e.g., LBFGS, which also guarantees global optimality and is often very fast.  See http://www.leemon.com/papers/1995b.pdf .  An empirical comparison is necessary.

**Experience Assessment:**

I have read many papers in this area.

**Review Assessment: Checking Correctness Of Derivations And Theory:**

I assessed the sensibility of the derivations and theory.

**Review Assessment: Checking Correctness Of Experiments:**

I assessed the sensibility of the experiments.

**Review Assessment: Thoroughness In Paper Reading:**

I read the paper at least twice and used my best judgement in assessing the paper.

---

> ### Author Response · Authors · 2019-11-15
> **Response to Reviewer #1**
>
> We thank the referees for their careful consideration of the paper.  We endeavor to answer the referees’ questions in this response; the suggestions for improvements are greatly appreciated and will be taken into account in any revision of the paper.
>
> 1. How restrictive the assumptions are in Definition 4.4?  In particular, R_min is defined from Assumption 2.2 as “the immediate reward … is bounded between [R_min, R_max]”.  So if we set R_min to negative infinity, the right-hand side of Eq 6 will be infinity, and so the condition is always met.  Is this really true?  At least, does the experiment satisfy Definition 4.4?
>
> Definition 4.4 (Global Convergence Condition) is a general theoretical condition for the global convergence for dynamic discrete choice model under unobserved heterogeneity. It mainly characterize the relation between the Lipschitz-smoothness, range of reward and the discount factor. For a specific MDP (i.e., Rust model studied in experiment section), directly using Def 4.4 will ensure the convergence for discount factor less than 0.5 without any additional restrictions on the utility model. However, the guarantee can improved with some additional knowledge of the structure of the model, or the optimal policy. In experiment section, since the optimal policy in Rust model is monotone, by further restricting to policy space of all monotone policies, our experiment suggests that the convergence happen even for discount factor equal 0.9 or 0.99.
>
> In this paper, we assume the immediate reward is lower bounded by some non-negative value R_min, which is necessary for the characterization between the conditional choice probability delta and threshold function pi. To our knowledge, all empirical papers in Economics and Marketing use dynamic discrete choice models that satisfy this restriction.
>
>
> 2. The experiment is on a relatively small problem.  Solving value/policy iteration with 2571 states and 2 actions is really not so hard, and many efficient algorithms exist other than value/policy iteration.  For example, a variant of policy iteration where policy evaluation is not solved exactly, but instead approximated by applying a small number of Bellman iterations. Or directly optimize the Bellman residual by, e.g., LBFGS, which also guarantees global optimality and is often very fast.  See http://www.leemon.com/papers/1995b.pdf .  An empirical comparison is necessary.
>
> We realize that Rust’s model is not the most impressive model in terms of scale. However, it is the standard benchmark used in theoretical work on dynamic discrete choice models in Economics and Marketing. One of our goals is to demonstrate excellent theoretical properties of the policy gradient for these types of discrete choice problems  to researchers in Economics and encourage them to start adopting the methods from RL. We run the experiment on Rust model as a sanity check for our theoretical results. We didn’t directly compare gradient descent method with other methods in this experiment. However, a conceptual advantage of policy gradient for problems studied in Econometrics can be thought of as follows.
> The literature of Econometric models for dynamic Economic behavior considers the optimization of individuals or small firms for whom the transformation probability in MDP is unknown and required learning. Due to the uncertainty of transformation probability, the value function iteration / policy iteration methods requires the construction of empirical MDP (which might be impractical for individuals or small firms), while the stochastic policy gradient does not require estimation of such probability.

---

### Official Review · AnonReviewer2 · 2019-10-23
**Official Blind Review #2**

**Rating:** 6

**Review:**

The paper is consider dynamic discrete choice models. It shows in an important class of discrete choice models the value function is globally concave in the policy, implying that for example policy gradients are globally convergent and are likely to converge faster in practice compared to fix-point approaches.

The paper is very well written and structured. It present convergence results together with a sanity check implementation. However, as an informed outsider, I am also a little bit confused. As far as I understand, Rust (1987) is also using gradient descent. Indeed the problem might not be convex/concave and hence this might get trapped in local minima. Moreover, Ermon et al. (AAAI 2015) have already shown that Dynamic Discrete Choice models are equivalent to Maximum Entropy IRL models under some conditions. Then they provide an algorithm that is kind of close (at least in spirit) to policy gradient. The propose to "simultaneously update the current parameter estimate θ (Learning) while we iterate over time steps t to fill columns of the DP table (Planning)". Indeed, this is still not giving guarantees, but the together with Ho et al. (ICML 2016) it suggests that the take-away message "use police gradients" for dynamic discrete choice models is actually known in the literature. This should bee clarified. Generally, the paper is providing a lot of focus on the economics literature. While this is of course fine, the authors should clarify what is already known in the AI and ML literature (including the work described above).

Nevertheless, the proof that there are convergent policy gradients for some dynamic discrete choice models appears interesting, at least to an informed outsider. However, this results heavily hinges on e.g. (Pirotta 2015). So the main novelty seems to be in Sections 4.3 and 4.4. Here is where they make use of their assumption. So, the only point, in my opinion, that should be clarified is the usefulness of the considered class. For an informed outsider, this is not easy to see.

**Experience Assessment:**

I have read many papers in this area.

**Review Assessment: Checking Correctness Of Derivations And Theory:**

I assessed the sensibility of the derivations and theory.

**Review Assessment: Checking Correctness Of Experiments:**

I assessed the sensibility of the experiments.

**Review Assessment: Thoroughness In Paper Reading:**

I read the paper at least twice and used my best judgement in assessing the paper.

---

> ### Author Response · Authors · 2019-11-15
> **Response to Reviewer #2**
>
> We thank the referees for their careful consideration of the paper.  We endeavor to answer the referees’ questions in this response; the suggestions for improvements are greatly appreciated and will be taken into account in any revision of the paper.
>
> 1. Rust (1987) is also using gradient descent. Indeed the problem might not be convex/concave and hence this might get trapped in local minima.
>
> Rust’s algorithms is the “nested fixed point” algorithm. For optimal policy he uses the standard fixed point algorithm (iterating for the value function). Then for each value function that comes out as a fixed point, he updates the vector of parameters of the likelihood function. This approach is known to be suffering from extremely slow performance in the Economics literature especially when the state space becomes larger. Our main theoretical contribution is the demonstration of concavity of the value function with respect to policy allowing us to use a non-fixed point gradient methods. It is absolutely true that for generic dynamic discrete choice models the problem might not be concave. However, as our results shown, assuming unobserved heterogeneity (i.e. random choice-specific shocks) follows Type I Extreme Value distribution, it induces a nice characterization of the conditional choice probability (i.e., softmax), the concavity of the problem is guaranteed.
>
> 2. Clarify what is already known in the AI and ML literature.
>
> Thanks for your comment, we will add the references to AI and ML literature into introduction section. As we briefly mentioned, the existing results on the global performance of the gradient-based algorithms for solving dynamic discrete choice problems is sparse.
>
> 3. Clarify the usefulness of the considered class.
>
> In our paper we consider a concrete model that is used for Econometric analysis of dynamic discrete choice in Economics and Marketing and provide global convergence guarantee for the policy gradient algorithm for that model. We reviewed theoretical papers studying dynamic discrete choice that came out in Econometrica (one of the top Economics journals) and all of them use Rust’s setup as a benchmark. The reason why we use Rust model is because most recent Econometric papers on single-agent dynamic decision-making uses Rust's setup to showcase their results.

---

### Official Review · AnonReviewer3 · 2019-11-04
**Official Blind Review #3**

**Rating:** 3

**Review:**

This paper deals with a certain class of models, known as discrete choice models. These models are popular in econometrics, and aim at modelling the complex behavioural patterns of individuals or firms. Entities in these models are typically modelled as rational agents, that behave optimally for reaching their goal of maximizing a certain objective function such as maximizing expected cumulative discounted payoff over a fixed period.

This class of models, modelled as a MDP with choice-specific heterogeneity, is challenging as not all the payoffs received by the agents is externally observable. One solution in this case is finding a balance condition and a functional fixed point to find the optimal policy (closely related to value function iteration), and this is apparently the key idea behind ‘nested fixed point’ methods used in Econometrics.

The paper proposes an alternative. First it identifies a subclass of discrete choice models (essentially MDP with stochastic rewards) where the value function is globally concave in the policy. The consequence of this observation is that a direct method, such as the policy gradient that circumvents explicitly estimating the value function, can (at least in principle) converge to the optimal policy without calculating a fixed point. The authors illustrate computational advantages of this direct approach. Moreover, the generality of the policy gradient method enables the relaxation of extra assumptions regarding the behaviour of agents while facilitating a wider applicability/econometric analysis.

The key contribution claimed by the paper is the observation that in the class of dynamic discrete choice models with unobserved heterogeneity, the value function is globally concave in the policy. This enables using computationally efficient policy gradient algorithms with convergence guarantees for this class of problems. The authors also claim that the simplicity of policy gradient makes it also a viable model for understanding economic behaviour in econometric analysis, and more broadly for social sciences.

The paper deals with Discrete choice models with unobserved heterogeneity as a special class of MDP’s and is relevant to ICLR. However, the writing style is quite technical and terse -- while I could appreciate the rigour, the authors develop the basic material until page 5 -- Notation is also somewhat non-standard at places (alternating using delta or sigma for a policy and pi for thresholds of an exponential softmax distribution) and makes it harder to see the additional structure from generic MDPs more familiar in RL. I suspect that a reader more familiar with the relevant econometric, marketing and finance literature could follow the model description more easily.

There are a number of assumptions in the paper, especially 2.4 and 2.5 that relate to the monotonicity and ordering of the states. These assumptions seem to be important in subsequent developments for showing the concavity but they seem to be coming from out of the blue. Unfortunately the authors do not provide any intuition/discussion -- an example problem with these properties would make these assumptions more concrete. I was hoping to find such an example in the empirical application however this section does not make the necessary connections with the theoretical development. There are not even references to basic claims
done in the abstract, for example, I am not able to find an illustration of ‘significant computational advantages in using a simple implementation policy gradient algorithm over existing “nested fixed point” algorithms used in Econometrics’. The lack of any conclusions makes it also hard for me to appreciate the contributions.


Minor:

Abstract:
.. Existing work in Econometrics assumes that optimizing agents are fully rational and requires finding a functional fixed point to find the optimal policy. …

Ambiguous sentence: Existing work in Econometrics [...] requires finding a functional fixed point to find the optimal policy.

Theorem 3.1 and elsewhere  Freéchet =>  Fréchet


**Experience Assessment:**

I have read many papers in this area.

**Review Assessment: Checking Correctness Of Derivations And Theory:**

I assessed the sensibility of the derivations and theory.

**Review Assessment: Checking Correctness Of Experiments:**

I assessed the sensibility of the experiments.

**Review Assessment: Thoroughness In Paper Reading:**

I read the paper at least twice and used my best judgement in assessing the paper.

---

> ### Author Response · Authors · 2019-11-15
> **Response to Reviewer #3**
>
> We thank the referees for their careful consideration of the paper.  We endeavor to answer the referees’ questions in this response; the suggestions for improvements are greatly appreciated and will be taken into account in any revision of the paper.
>
> 1. Intuition and discussion for Assumption 2.4, 2.5.
>
> Assumption 2.4 implies that taking one action results in next-period state distribution that FOSD the next-period state distribution if the other action is taken. This assumption would be satisfied in multiple economic setting such as, for example:
> The bus engine replacement. Replacing the engine today implies that it is less likely that by the end of tomorrow the bus mileage will be close to the maximum (think of the state as the negative mileage).
> Choice about schooling studied in Card (Econometrica, 2001). Getting a college degree increases the probability of getting a higher salary.
> Behavior of consumers making purchasing decisions with respect to storable goods studied in Hendel and Nevo (Econometrica, 2006). Purchasing a good today implies lower probability of having low inventories (and possibly running out of) the good tomorrow.
>
> Assumption 2.5 suggests monotonicity of preferences of the decision maker over states. That is, higher values of the state are associated with higher utility, which would be true in all of the situations described above: the larger is the difference between the maximum mileage and the current mileage, the cheaper it is to maintain the bus; the larger the salary/the higher the inventory, the better off the person is.
>
>
> 2. The illustration of ‘significant computational advantages in using a simple implementation policy gradient algorithm over existing “nested fixed point” algorithms used in Econometrics’.
>
> Please see the response of comment 2 in Reviewer 1.

---

### Decision · Program_Chairs · 2019-12-19

**Decision:**

Reject

**Comment:**

The authors develop theoretical results showing that policy gradient methods converge to the globally optimal policy for a class of MDPs arising in econometrics. The authors show empirically that their methods perform on a standard benchmark.

The paper contains interesting theoretical results. However, the reviewers were concerned about some aspects:
1) The paper does not explain to a general ML audience the significance of the models considered in the paper - where do these arise in practical applications? Further, the experiments are also limited to a small MDP - while this may be a standard benchmark in econometrics, it would be good to study the algorithm's scaling properties to larger models as is standard practice in RL.

2) The implications of the assumptions made in the paper are not explained clearly, nor are the relative improvements of the authors' work relative to prior work. In particular, one reviewer was concerned that the assumptions could be trivially satisfied and the authors' rebuttal did not clarify this sufficiently.

Thus, I recommend rejection but am unsure since none of the reviewers nor I am an expert in this area.